# Immunogenicity, Safety, and Efficacy of Influenza Vaccine in T2DM and T2DM with Chronic Kidney Disease

**DOI:** 10.3390/vaccines12030227

**Published:** 2024-02-23

**Authors:** Henhen Heryaman, Cep Juli, Arnengsih Nazir, Mas Rizky A. A. Syamsunarno, Badrul Hisham Yahaya, Dewi Kartika Turbawaty, Rini Mulia Sari, Hikmat Permana, Rudi Supriyadi, Nur Atik

**Affiliations:** 1Doctoral Program, Faculty of Medicine, Padjadjaran University, Bandung 40161, Indonesia; henhen@unpad.ac.id (H.H.); arnengsih@unpad.ac.id (A.N.); 2Department of Biomedical Sciences, Faculty of Medicine, Padjadjaran University, Bandung 45363, Indonesia; rizky@unpad.ac.id; 3Department of Neurology, Dr. Hasan Sadikin General Hospital, Faculty of Medicine, Padjadjaran University, Bandung 40161, Indonesia; cepjuli42@gmail.com; 4Department of Biomedical Sciences, Advanced Medical and Dental Institute (IPPT), Universiti Sains Malaysia, Bertam Kepala Batas 13200, Malaysia; badrul@usm.my; 5Departemen of Clinical Pathology, Dr. Hasan Sadikin General Hospital, Faculty of Medicine, Universitas Padjadjaran, Bandung 40161, Indonesia; dewi.kartika@unpad.ac.id; 6PT Biofarma, Bandung 40161, Indonesia; rini.mulia@biofarma.co.id; 7Division of Endocrinology and Metabolism, Internal Medicine Department, Dr. Hasan Sadikin General Hospital, Faculty of Medicine, Universitas Padjadjaran, Bandung 40161, Indonesia; hikmat.permana@unpad.ac.id; 8Division of Nephrology, Internal Medicine Department, Dr. Hasan Sadikin General Hospital, Faculty of Medicine, Universitas Padjadjaran, Bandung 40161, Indonesia; rudi.supriyadi@unpad.ac.id

**Keywords:** clinical trial, immunogenicity, trivalent influenza vaccine, T2DM, T2DM–CKD

## Abstract

Patients with Type 2 diabetes mellitus (T2DM) and Chronic Kidney Disease (CKD) face an increased risk of morbidity and mortality after influenza infection. Several studies have shown that the influenza vaccine effectively prevents morbidity and mortality in T2DM patients. However, there has been limited research aimed at assessing the effectiveness of the trivalent influenza vaccine in T2DM–CKD patients. This study aimed to identify Geometric Mean Titers (GMTs), seroprotection, seroconversion, safety, and efficacy. This open-label clinical trial was conducted at AMC Hospital in Bandung, West Java, Indonesia between June 2021 and July 2022. The study subjects consisted of 41 T2DM and 26 T2DM–CKD patients who were administered the trivalent influenza vaccine. There was a significant difference in the average age, with the T2DM–CKD patients being older. Median titers post-vaccination for the B/Washington virus were higher in the T2DM patients compared to the T2DM–CKD patients, and this difference was statistically significant. A majority, comprising 75.6% of the T2DM and 80.8% of the T2DM–CKD patients monitored post-influenza-vaccination, did not experience any adverse reactions. The most common reaction was the sensation of fever, with incidence rates of 12.2% in the T2DM patients and 15.4% in the T2DM–CKD patients. Furthermore, we observed that the incidence of Influenza-like Illness was highest at 7.3% in the T2DM patients and 7.7% in the T2DM–CKD patients. The trivalent influenza vaccine demonstrated equivalent safety and effectiveness in both groups.

## 1. Introduction

Seasonal influenza is a sudden-onset respiratory infection resulting from the activity of influenza viruses that are prevalent across the globe, and it remains a prevalent public health issue. Influenza A and B viruses actively circulate and contribute to the annual occurrence of seasonal disease epidemics. An influenza virus infects the upper respiratory tract, causing mild flu-like symptoms that can progress to more severe manifestations. The annual number of severe influenza cases ranges from 3 to 5 million, resulting in 290,000 to 650,000 deaths from respiratory causes alone [1,2]. Influenza can also lead to direct infection in the form of lower respiratory tract infection and lead to death, ranging from 99,000 to 200,000 cases [3]. Individuals aged over 60 years have a higher risk of mortality compared to younger age groups [4]. Individuals diagnosed with diabetes mellitus (DM) and those vulnerable to infections like renal disease who acquire influenza face an increased likelihood of developing lower respiratory tract infections including pneumonia [5].

Additionally, a previous study indicated that DM patients infected with the influenza A (H1N1) virus have a three-fold risk of hospitalization and a four-fold risk of requiring treatment in an Intensive Care Unit (ICU) [6]. Individuals with CKD experience a compromised immune response, making their immune systems less capable of combating infections. Consequently, all stages of CKD face an increased risk of severe illness due to influenza infection [7].

The primary means of preventing the disease is vaccination, with safe and effective vaccines existing for over six decades. However, it is crucial to acknowledge that immunity acquired through vaccination wanes over time. Administering a single dose of the trivalent influenza vaccine in healthy adults can provide the same level of protection after 3 months as in adults who have regularly been vaccinated every year [8]. Individuals with T2DM have a reduced efficacy toward the influenza vaccine; however, other studies have shown that the immunogenicity of the influenza vaccine in individuals with T2DM is generally comparable to that in healthy individuals [9,10]. Therefore, annual vaccinations are recommended to maintain continuous protection against influenza [1]. The aim of this study was to discern discrepancies in the efficacy, safety, and immunogenicity of the influenza vaccine between individuals with Type 2 diabetes mellitus (T2DM) and those with Type 2 diabetes accompanied by Chronic Kidney Disease (T2DM–CKD).

## 2. Materials and Methods

### 2.1. Study Design and Participant

The study was conducted as an open-label clinical trial designed between June 2021 and July 2022 at the outpatient department of AMC Hospital in Bandung, West Java Province, Indonesia. Forty-one subjects with T2DM and twenty-six subjects with T2DM with CKD were invited to participate in this study.

The inclusion criteria mandated that participants possessed T2DM and fell within the age range of 40 to 59. Prior to inclusion in the study, informed consent was obtained from all subjects. Those with elevated fever (>38 °C), convulsions, or acute infections were excluded from participation. Additional exclusion criteria encompassed a history of egg allergy or hypersensitivity to the influenza vaccine, CKD Stage V, pregnancy, ongoing long-term use of steroids, presence of malignancy, autoimmune diseases, a documented history of SARS-CoV-2 infection evidenced by a negative PCR test result within the past 14 days, and the receipt of a COVID-19 vaccine within the previous 14 days. Blood samples were collected from all participants who were enrolled in the study to evaluate antibody titers associated with the influenza vaccine, assess safety and efficacy, and investigate the occurrence of Influenza-like Illness related to the influenza vaccine one month post-administration. The study was registered on ClincalTrials.gov (NCT06252051).

### 2.2. Ethical Approval

The research protocol obtained approval from the Research Ethics Board of Padjadjaran University No. 257/UN6.KEP/EC/2020.

### 2.3. Data Collection and Study Intervention

Clinical data information was obtained from medical records and interviews after obtaining informed consent. The interviews were conducted to gather information about the duration of DM, history of influenza vaccination, history of COVID-19 vaccination, egg allergy history, history of steroid usage, and malignancy history. The researchers also collected basic demographic data such as age, gender, routine medications, and pre-existing medical conditions from the medical records. For other anthropometric data such as BMI, Waist Circumference, Total Body Fat, and Visceral Fat content, measurements were taken shortly before administering the influenza vaccine. Laboratory examinations included fasting blood glucose, lipid profile, creatinine, routine urine tests, and influenza antibody titers. Fasting blood glucose and influenza antibody tests were conducted twice—before and after vaccine administration—with an interval of 28–35 days after vaccination.

T2DM is defined by a fasting plasma glucose level greater than 126 mg/dL (7.0 mmol/L), HbA1c of 6.5 or higher, plasma glucose levels exceeding 200 mg/dL 2 h after an oral glucose tolerance test, or random plasma glucose levels surpassing 200 mg/dL in patients exhibiting classic symptoms. Glycemic index is controlled if fasting glucose level < 126 mg/dL [11]. BMI classification follows the Asia–Pacific criteria: BMI below 18.5 is categorized as underweight, 18.5–22.9 as normal, 23–24.9 as overweight, 25–29.9 as pre-obese, and 30 or higher as obese [12]. 

Chronic Kidney Disease is indicated by a decrease in glomerular filtration rate to less than 60 mL/minute/1.73 m^2^ for a minimum of 3 months [13]. Cholesterol classification follows ATP (Adult Treatment Panel III) definition for LDL (Low-Density Lipoprotein) Cholesterol, where <100 is Optimal, 100–129 is Near-Optimal/Above-Optimal, 130–159 is Borderline-High, 160–189 is High, and ≥190 is Very High. For Total Cholesterol, <200 is Desirable, 200–239 is Borderline-High, and ≥240 is high. For HDL (High-Density Lipoprotein) Cholesterol, <40 is Low and ≥60 is High [14]. Staging CKD follows Estimated Glomerular Filtration Rate (eGFR) classification CKD EPI 2021 [15,16]. Blood tests for glucose, lipid profile, urine, and creatinine were conducted in the Clinical Pathology laboratory of AMC Hospital. The assessment of body fat percentage and visceral fat was conducted using Body Impedance Analysis (BIA). For men, body fat percentage was classified as follows: <10% as low, 10–20% as normal, 20–25% as high, and >25% as very high. For women, the classification for body fat was as follows: <20% as low, 20–30% as normal, 30–35% as high, and >35% as very high. Visceral fat classification was as follows: 0.5–9.5 as normal (0), 10.0–14.5 as high (+), and 15.0–30.0 as very high (++) [17]. The classification of waist circumference according to the World Health Organization (WHO) is based on gender. For men, <90 cm is considered “desirable” while >90 cm is classified as “undesirable”. For women, <80 cm is considered “desirable” and >80 cm is classified as “undesirable” [12].

In this study, the influenza vaccine used was formulated by Biofarma, Bandung, Indonesia and had the lot number 3,020,221. Each 0.5 mL dose of Flubio contained the following virus strains: B/Washington/02/2019-like virus (B Victoria/705/2018, BVR-11), A/Hong Kong/2671/2019 (H3N2)-like virus (A/Hong Kong/2671/2019, NIB-121), and A/Guangdong-Maonan/SWL1536/2019 (H1N1)PDM09-like virus (A/Guangdong-Maonan/SWL1536/2019, CNIC-1909).

The vaccine was stored at temperatures ranging from 2 to 8 °C. Administration of the vaccine was performed via intramuscular injection in the left deltoid using a 24-gauge needle, with each subject receiving a 0.5 mL dose (1 dose). Blood samples of 5 mL were collected in vacutainer tubes from all groups before vaccination and at the second visit (V1) after the vaccine, with intervals of 28 to 35 days. All samples were properly labelled and stored at −80 °C in the Faculty of Pharmacy, Padjadjaran University. The antibody titers were measured using the hemagglutination inhibition (HI) test in the Immunology Laboratory of the Clinical Trial Department at Biofarma.

### 2.4. Geometric Means Titers and Clinical Outcome

The objective of this analysis was to compare the immunogenicity among the three groups by calculating the Geometric Mean Titers (GMTs) and seroconversion and seroprotection rates. The average increase in GMT before and after vaccine administration within a period of 28–35 days was also assessed.

We defined seroconversion as having antibody titers of at least 1:40 and a four-fold or greater increase in hemagglutination inhibition (HI) titers following vaccination. We recorded subjects from both groups who had antibody titers > 1:40 one month after receiving the vaccine.

To monitor symptoms of flu-like illness, each participant received an observation card and was instructed to document any experienced symptoms within the 28–35 days post-vaccination. At the second meeting, held within the same timeframe, observation cards were collected and fasting blood glucose levels were analyzed. This facilitated the assessment of potential correlations between vaccine administration and alterations in fasting blood glucose levels.

### 2.5. Statistical Analysis

The clinical and demographic characteristics of the enrolled participants were detailed through frequency distribution tabulation. Influenza antibody levels for each strain were presented as geometric means, along with a 95% confidence interval (CI). All the data were analyzed for distribution before proceeding to proper statistical analysis. To assess differences between numerical and categorical variables across groups, the Mann–Whitney test, *t*-test, and Chi-square test were employed. The Kruskal–Wallis test was utilized to compare the median range of titers, and seroprotection and seroconversion rates for each influenza virus strain in both groups were compared using the Fisher’s exact test. The occurrence of Influenza-Like Illness (ILI) and efficacy one month after vaccine administration were also described, with the analysis employing computation of numbers and percentages. All data were analyzed using Statistical Product and Service Solution (SPSS) software, version 25.0, for Windows (IBM Corp., Armonk, NY, USA) and GraphPad PRISM, version 9.

## 3. Result 

The T2DM group comprised 41 participants while the T2DM–CKD group included 26 participants in this study. Predominantly, both groups had a majority of female participants, accounting for 77.1% in the T2DM group and 73.1% in the T2DM–CKD group. No statistically significant difference in gender distribution between the two groups was observed, with a *p*-value of 0.642 (Table 1).

The average age of participants in the T2DM–CKD group (53.2 years) was notably higher than that in the T2DM group (49.8 years), and this difference was statistically significant with a *p*-value of 0.009. Although the distribution of participants across specific age categories varied between the two groups, these differences were not statistically significant (*p*-value > 0.05). Duration of diabetes did not exhibit a significant difference between the groups, as evidenced by a *p*-value of 0.687. Similarly, the distribution of participants across duration of diabetes categories showed no significant difference between the two groups (Table 1).

Analyses of total body fat percentage and waist circumference in both males and females revealed no significant differences between the two groups. Visceral fat also did not differ significantly between the T2DM and T2DM–CKD groups. Consequently, the primary conclusion drawn from these baseline characteristics (Table 1) was the significant age difference between the two groups, with the T2DM–CKD group displaying a higher average age.

The average fasting glucose levels before vaccination were nearly identical between the two groups, at around 155 mg/dL, and the difference was not statistically significant. Similarly, the average fasting glucose levels after vaccination showed no substantial disparity between the two groups, with approximately 160 mg/dL for the T2DM group and 163 mg/dL for the T2DM–CKD group (Table 2).

We also found that there were no significant differences in lipid profiles between the two groups. This included total cholesterol, LDL (Low-Density Lipoprotein), HDL (High-Density Lipoprotein), and triglycerides (Table 2).

Interestingly, a significant difference in eGFR was observed between the two groups. The T2DM group exhibited a higher average eGFR (108 mL/min/1.73 M^2^) compared to the T2DM–CKD group (61 mL/min/1.73 M^2^). The distribution of participants in eGFR categories also revealed a highly significant difference between the two groups. The T2DM group comprised all participants with an eGFR above 90 mL/min/1.73 M^2^ while the T2DM–CKD group had the majority of participants with an eGFR below 60 mL/min/1.73 M^2^ (Table 2). 

The results indicate that there was a significant difference in kidney function (eGFR) between the two groups, with the T2DM–CKD group having lower eGFR values. However, there were no significant differences in fasting glucose parameters before or after vaccination or in lipid profiles between the two groups.

We then analyzed the antibody levels before and after vaccination against three types of influenza viruses (B/Washington, A/Hong Kong (H3N2), and A/Guangdong (H1N1)) for two groups: T2DM (Type 2 Diabetes) and T2DM–CKD (Type 2 Diabetes with Kidney Disease) (Table 3).

No GMT data could be calculated for both the T2DM and T2DM–CKD groups for the B/Washington virus because the antibody titers of some subjects were below the detection limit (0). However, for the A/Hong Kong (H3N2) and A/Guangdong (H1N1) viruses, there were no significant differences between the two groups. Median antibody titers for the A/Hong Kong (H3N2) and A/Guangdong (H1N1) viruses were similar between the two groups before vaccination. For the B/Washington virus, the median antibody titer in the T2DM–CKD group was lower, but this difference was not statistically significant. The percentage of participants with antibody titers of at least 40 (indicating seroprotection) did not show a significant difference between the two groups for all three types of viruses (Table 3).

One month after vaccination, the GMT for the B/Washington virus was higher in the T2DM group compared to the T2DM–CKD group, and this difference was statistically significant (*p* < 0.05). However, for the A/Hong Kong (H3N2) and A/Guangdong (H1N1) viruses, the difference between the two groups was not statistically significant. After vaccination, the median antibody titer for the B/Washington virus was higher in the T2DM group compared to the T2DM–CKD group, and this difference was statistically significant (*p* < 0.05). However, there was no statistically significant difference between the two groups for the A/Hong Kong (H3N2) and A/Guangdong (H1N1) viruses (Table 3).

The percentage of participants with antibody titers of at least 40 (indicating seroprotection) did not show a significant difference between the two groups for all three types of viruses. We further analyzed the Median Fold Titers and compared antibody titers after vaccination to those before vaccination. A significant difference was observed for the B/Washington virus wherein the T2DM group demonstrated a larger increase in antibody titers compared to the T2DM–CKD group (*p* < 0.05). However, there were no significant differences between the two groups for the A/Hong Kong (H3N2) and A/Guangdong (H1N1) viruses (Table 3).

Influenza vaccination appeared to be effective in generating good seroprotection against influenza viruses in both groups. Although there were differences in antibody levels between the two groups after vaccination, this difference was statistically significant only for the B/Washington virus, with the T2DM group showing a greater increase in titers. There were no significant differences between the two groups for the A/Hong Kong (H3N2) and A/Guangdong (H1N1) viruses.

Further, we analyzed antibody responses to the B/Washington virus before and after vaccination in three groups of patients with Chronic Kidney Disease (CKD) at three different levels: CKD Stage 2, CKD Stage 3A, and CKD Stage 3B (Table 4). 

No GMT data could be calculated for CKD Stage 2 because the antibody titers of some subjects were below the detection limit (0). For CKD Stage 3A and CKD Stage 3B, the GMT values before vaccination were similar, at around 11. The median antibody titer before vaccination did not show a significant difference among the three CKD patient groups. The median antibody titer was 10 for all groups (Table 4).

One month after vaccination, the GMT for the B/Washington virus was higher in CKD Stage 3A patients (90) compared to CKD Stage 2 (76) and CKD Stage 3B (40) patients. However, this difference was not statistically significant. The median antibody titer after vaccination did not show a significant difference among the three CKD patient groups. The percentage of patients with antibody titers of at least 40 (indicating seroprotection) after vaccination did not show a significant difference among the three CKD patient groups. The percentage of patients who experienced a change in antibody titers from before vaccination to titers of at least 40 after vaccination also did not show a significant difference among the three CKD patient groups. Median Fold Titers did not show a significant difference among the three CKD patient groups (Table 4).

We found that antibody responses to the B/Washington virus in the influenza vaccine seem to generate similar antibody responses before vaccination among patients at different stages of CKD. Following vaccination, there is a trend toward an increased GMT in CKD Stage 3A patients, but this difference is not statistically significant. Antibody responses to influenza vaccination in CKD patients appear to be consistent.

We also analyzed antibody responses to the A/Hong Kong (H3N2) virus before and after vaccination in three groups of patients with CKD at three different levels: CKD Stage 2, CKD Stage 3A, and CKD Stage 3B (Table 4).

Before vaccination, the GMT antibody titer for the A/Hong Kong (H3N2) virus was higher in CKD Stage 3A patients (90) compared to CKD Stage 2 (46) and CKD Stage 3B (53) patients. However, this difference was not statistically significant. The median antibody titer before vaccination did not show a significant difference among the three CKD patient groups. The percentage of patients with antibody titers of at least 40 before vaccination did not show a significant difference among the three CKD patient groups (Table 4).

After vaccination, the GMT for the A/Hong Kong (H3N2) virus was higher in CKD Stage 3A patients (285) compared to CKD Stage 2 (263) and CKD Stage 3B (184) patients. However, this difference was not statistically significant. The median antibody titer after vaccination did not show a significant difference among the three CKD patient groups. The percentage of patients with antibody titers of at least 40 after vaccination was high (100%) and did not show a significant difference among the three CKD patient groups. The percentage of patients who experienced seroconversion after vaccination did not show a significant difference among the three CKD patient groups. Median Fold Titers did not show a significant difference among the three CKD patient groups (Table 4).

In our investigation, we observed that the A/Hong Kong virus efficiently triggers antibody responses before influenza vaccine administration across different CKD stages, with CKD Stage 3A exhibiting the highest Geometric Mean Titers. One month after vaccination, the antibody response to the A/Hong Kong virus is comparable to that of the B/Washington virus. While the GMT of the A/Hong Kong virus is higher than that of the B/Washington virus, the Median Fold Titers of the B/Washington virus demonstrate a larger increase compared to the A/Hong Kong virus (Table 4).

We additionally examined antibody responses to the A/Guangdong virus both before and after vaccination in three patient groups with CKD at different stages: CKD Stage 2, CKD Stage 3A, and CKD Stage 3B (Table 4).

The initial A/Guangdong virus GMT antibody titers before vaccination were relatively similar among the three CKD patient groups, i.e., those for CKD Stage 2 (59), CKD Stage 3A (64), and CKD Stage 3B (61). The median antibody titer before vaccination did not show a significant difference among the three CKD patient groups. The percentage of patients with antibody titers of at least 40 before vaccination did not show a significant difference among the three CKD patient groups (Table 4).

Additionally, the GMT after vaccination remained uniform among the three CKD patient groups: those for CKD Stage 2 (905), CKD Stage 3A (905), and CKD Stage 3B (1114). There were no significant differences between these groups. The median antibody titer after vaccination did not show a significant difference among the three CKD patient groups. The percentage of patients with antibody titers of at least 40 after vaccination was high (100%) and did not show a significant difference among the three CKD patient groups. The percentage of patients who experienced seroconversion after vaccination also did not show a significant difference among the three CKD patient groups. Median Fold Titers did not show a significant difference among the three CKD patient groups (Table 4).

We found that the three virus strains present in the influenza vaccine exhibit similar degrees of effectiveness in eliciting antibody responses in the three CKD groups. The antibody response in CKD patients remains consistent after one month of vaccination, with no significant differences in GMT, median titers, seroprotection, or seroconversion observed among the three CKD patient groups. The median titers, GMT, and Median Fold Titers for A/Guangdong virus were the highest compared to the B/Washington virus and A/Hong Kong virus (Table 4).

Out of the 41 subjects with T2DM who were monitored for post-vaccination reactions, 31 of them (75.60%) did not experience any reactions after vaccination (Figure 1A). Similarly, in the T2DM–CKD group, 21 subjects (80.76%) did not report any reactions following vaccination. The most commonly reported reaction was fever among the T2DM subjects, with an incidence rate of 5 cases (12.19%), while in the DM–CKD group, 4 cases (15.38%) experienced fever.

The second most frequently observed post-vaccination reaction was pain at the injection site, which occurred in three cases (7.32%) among T2DM patients. Additionally, diarrhea was noted as a post-vaccination reaction, with two cases (4.88%) in the T2DM group and one case (3.85%) in the DM–CKD group. Based on these occurrences of post-vaccination reactions, it can be concluded that the influenza vaccine is safe for individuals with T2DM and T2DM–CKD.

Influenza-like Illness (ILI) monitoring (Figure 1B) was conducted to assess the effectiveness of the influenza vaccine within 1 month in both groups. The symptoms resulting from influenza virus infection are almost the same as those from the common cold, including cough, sore throat, runny nose, fever, headache, fatigue, and weakness. However, influenza patients tend to have more pronounced symptoms, especially in terms of fever, pain, fatigue, weakness, and headache. Fever was the most common event in both groups one month after receiving the vaccine. In the T2DM group, there were three cases of fever (7.32%), while in the T2DM–CKD group, there were two cases (7.69%). Other ILI events were nearly the same, with only one subject experiencing cough–fever, cough–cold, and cough–cold–fever, with a percentage of 3.85% in the T2DM–CKD group and 2.44% in the T2DM group.

## 4. Discussion 

Influenza virus infection continues to pose a global health challenge, contributing to heightened morbidity and mortality rates, especially among vulnerable populations. Individuals with T2DM represent one such vulnerable group in the context of influenza susceptibility. Apart from their increased vulnerability to influenza, people with diabetes frequently encounter complications, particularly diabetic kidney disease (DKD), which may progress to CKD. Notably, CKD patients face an elevated risk of infections due to compromised immune system functionality.

Our study highlights a noteworthy finding: subjects with T2DM are generally younger than those with T2DM–CKD. It is widely recognized that the decline in kidney function correlates directly with age, with kidney function diminishing progressively as individuals age. Typically, there is an approximate 10% reduction in kidney function per decade from the age of 30 to 80. Moreover, age-related changes also impact nephron function, contributing to a roughly 10% decrease in kidney cortex thickness per decade as individuals age [18,19,20].

The target glycemic index in T2DM and DM–CKD is still low, being below 44%. This condition is almost similar to the glycemic target based on HbA1c (<7%), which is around 37% [21]. A glycemic target based on fasting glucose is more variable and easier to change compared to HbA1c levels, which is why the percentage is higher. Fasting blood glucose experiences rapid changes compared to HbA1c levels because fasting blood glucose is not like HbA1c, which undergoes glycation between glucose and hemoglobin and can therefore represent the average glucose level over the past 3 months. Fasting blood glucose tests have the advantage of being inexpensive and can be performed immediately, but they have the drawback of being influenced by acute illness and stress [22].

Further, we also found that the fasting blood glucose experiences an increase after vaccination. There are various conditions that can elevate fasting blood glucose levels post-vaccination, including medication adherence, controlled diet, physical activity, and the presence of an infection or inflammation caused by an influenza virus resulting in decreased systemic insulin sensitivity in the liver and muscles [23]. The administration of the influenza vaccine can lead to stress and acute illness. This condition leads to an increase in glucagon hormone levels, resulting in glycogenolysis and gluconeogenesis, leading to an increase in fasting blood sugar levels.

Most of the subjects with T2DM and T2DM–CKD did not exhibit immunity against the influenza virus strain B/Washington. Immunity against this strain, unlike strains A/Hong Kong and A/Guangdong, is typically observed when antibody titers are below 1:40 (seroprotection). The lack of immunity in this case was likely because the B/Washington strain rarely causes infections. Despite this, the World Health Organization (WHO) recommended it for vaccination in the Northern Hemisphere in 2021, including the B/Washington/02/2019 (B/Victoria lineage)-like virus [24].

Following vaccination, both the GMT and median titers for the B/Washington virus demonstrated a statistically significant increase in the T2DM group compared to the T2DM–CKD group. This emphasizes that despite DM potentially inducing immune dysregulation, affecting both cellular and humoral immune responses and potentially leading to impaired antibody production by B cells, the elevation in antibody titers against the B/Washington strain after vaccination was more pronounced in the T2DM group compared to the T2DM–CKD group.

DKD is defined as a form of CKD resulting from DM, with a prevalence of approximately 30–40% among individuals with diabetes [25]. Hyperglycemia in DM significantly influences the function of anti-viral immune cells. HbA1c levels demonstrate a positive correlation with the duration and severity of infection from various pathogens [26]. For over two decades, DKD has maintained its status as the primary cause of CKD, with inflammation and immune responses implicated in its development [27].

We acknowledge that our study has limitations, including the cellular mechanisms that could influence antibody titers in T2DM both with and without CKD. Vaccination also induces the development of memory cells specific to influenza in both CD4 and CD8 T-cells. These memory cells generate effectors that migrate to the lungs, contributing to the clearance of virus-infected cells in the pulmonary region [28]. Future studies should explore and analyze this factor in the context of T2DM with or without CKD.

## 5. Conclusions

In conclusion, the influenza vaccine elicited comparable immune responses in both the T2DM and T2DM–CKD groups. Notably, the immune response to the B/Washington virus was notably more robust in the T2DM group compared to the T2DM–CKD group. Overall, the trivalent influenza vaccine demonstrated a commendable safety profile and effectiveness, affording robust protection for both the T2DM and T2DM–CKD groups.

## Figures and Tables

**Figure 1 vaccines-12-00227-f001:**
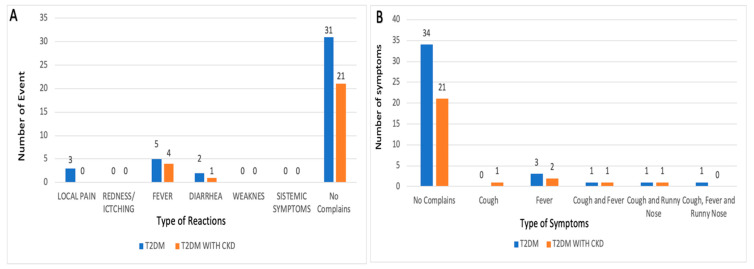
Reaction and Influenza Like Illness after influenza vaccine: (**A**) safety of influenza vaccine; (**B**) efficacy of influenza vaccine.

**Table 1 vaccines-12-00227-t001:** Baseline characteristics of participants.

Variable	T2DM (*n* = 41)	T2DM–CKD (*n* = 26)	*p*-Value
Sex, *n* (%)			
Female	32 (77.1)	19 (73.1)	*0.642* ^c^
Male	9 (21.9)	7 (26.9)	
Age (years), Mean ± SD	49.8 ± 5.4	53.2 ± 4.5	*0.009* ^a^*
Age Category, *n* (%)			
40–44 years	10 (24.4)	2 (7.7)	*0.118* ^c^
45–49 years	9 (21.9)	3 (11.5)	
50–54 years	12 (29.3)	9 (34.6)	
55–59 years	10 (24.4)	12 (46.2)	
Duration of DM (years), Mean ± SD	5.5 ± 5.2	4.9 ± 2.9	*0.687* ^b^
Duration of DM Category, *n* (%)			
<5 years	27 (65.9)	14 (53.8)	*0.326* ^c^
≥5 years	14 (34.1)	12 (46.2)	
BMI (kg/m^2^), Mean ± SD	28.5 ± 4.1	27.2 ± 3.9	*0.187* ^a^
BMI Category, *n* (%)			
Underweight	1 (2.4)	1 (3.9)	*0.720* ^c^
Normal	0 (0.0)	1 (3.9)	
Overweight	9 (21.9)	5 (19.2)	
Pre-obese	17 (41.5)	12 (46.2)	
Obese	14 (34.2)	7 (26.8)	
Total Fat Mass (%), Mean ± SD			
Female	36.8 ± 4.7	34.8 ± 3.6	*0.121* ^a^
Male	28.0 ± 4.9	24.9 ± 5.7	*0.256* ^a^
Total Visceral Fat, Mean ± SD	12.7 ± 5.9	12.0 ± 6.8	*0.528* ^a^
Waist Circumference (cm), Mean ± SD			
Female	93.7 ± 10.2	91.6 ± 9.3	*0.477* ^a^
Male	98.5 ± 10.8	92.8 ± 12.6	*0.345* ^a^

Note: ^a^ Unpaired *t*-test, ^b^ Mann–Whitney test, ^c^ Chi-square test, * Significance: *p* < 0.05.

**Table 2 vaccines-12-00227-t002:** Baseline characteristics of laboratory participants.

Variable	T2DM (*n* = 41)	T2DM–CKD (*n* = 26)	*p*-Value
Fasting Glucose Pre-Vaccine (mg/dL), Mean ± SD	155 ± 63	155 ± 82	*0.695* ^b^
Fasting Glucose Post-Vaccine (mg/dL), Mean ± SD	160 ± 53.79	163 ± 59	*0.969* ^b^
Lipid Profile			
Total Cholesterol (mg/dL), Mean ± SD	202 ± 47	215 ± 45	*0.288* ^a^
LDL (mg/dL), Mean ± SD	143 ± 43	148 ± 41	*0.677* ^a^
HDL (mg/dL), Mean ± SD	52 ± 12	52 ± 14	*0.983* ^a^
Triglyceride (mg/dL), Mean ± SD	205 ± 144	242 ± 171	*0.325* ^b^
eGFR (mL/min/1.73 M^2^), Mean ± SD	108 ± 8	61 ± 19	*<0.001* ^a^*
eGFR Category, *n* (%)			
≥90	41 (100)	0 (0)	*<0.001* ^c^*
60–89	0 (0)	14 (53.8)	
45–59	0 (0)	6 (23.1)	
30–44	0 (0)	5 (19.2)	
15–29	0 (0)	1 (3.9)	

Note: ^a^ Unpaired *t*-test, ^b^ Mann–Whitney test, ^c^ Chi-square test, * Significance: *p* < 0.05.

**Table 3 vaccines-12-00227-t003:** Antibody titers pre- and post-vaccination.

Time Period	B/Washington			A/Hong Kong (H3N2)			A/Guangdong (H1N1)		
	T2 DM (*n* = 41)	T2 DM with CKD (*n* = 26)	*p* Value	T2 DM (*n* = 41)	T2 DM with CKD (*n* = 26)	*p* Value	T2 DM (*n* = 41)	T2 DM with CKD (*n* = 26)	*p* Value
Pre-vaccination									
Geometric Mean Titers (95% CI)	-	-		65 (49–87)	55 (42–72)		56 (43–72)	60 (44–81)	
Median Titers (Range)	10 (0–40)	10 (0–40)	*0.965* ^a^	40 (20–1280)	40 (20–320)	*0.553* ^a^	80 (10–160)	40 (20–320)	*0.903* ^a^
Antibody Titer ≥ 40, *n* (%)	3 (7.3)	1 (3.8)	*1.000* ^c^	35 (85.4)	24 (92.3)	*0.469* ^c^	34 (82.9)	22 (84.6)	*1.000* ^c^
Post-vaccination (1 month)									
Geometric Mean Titers (95% CI)	118 (87–160)	66 (42–104)		392 (297–518)	252 (163–388)		629 (485–817)	747 (741–1167)	
Median Titers (Range)	160 (20–640)	80 (10–640)	*0.035* ^a^*	320 (80–1280)	320 (40–1280)	*0.109* ^a^	640 (40–1280)	1280 (320–5120)	*0.115* ^a^
Seroprotection, *n* (%)	38 (92.7)	20 (76.9)	*0.080* ^c^	41 (100.0)	26 (100.0)	-	41 (100.0)	26 (100.0)	*-*
Seroconversion, *n* (%)	40 (97.6)	23 (88.5)	*0.291* ^c^	33 (80.5)	16 (61.5)	*0.088* ^b^	37 (90.2)	24 (92.3)	*1.000* ^c^
Median Fold Titers (95% CI)	16 (8–32)	8 (4–16)	*0.029* ^a^*	4 (4–8)	4 (2–8)	*0.249* ^a^	8 (8–16)	16 (8–32)	*0.115* ^a^

Note: ^a^ Mann–Whitney test, ^b^ Chi-square test, ^c^ Fisher’s exact test, * Significance: *p* < 0.05.

**Table 4 vaccines-12-00227-t004:** Trivalent antibody responses pre- and post-vaccination for any stage of CKD.

Time Period	B/Washington				A/Hong Kong (H3N2)				A/Guangdong (H1N1)			
	CKD Stage 2 (*n* =14)	CKD Stage 3A (*n* = 6)	CKD Stage 3B (*n* = 5)	*p* Value	CKD Stage 2 (*n* =14)	CKD Stage 3A (*n* = 6)	CKD Stage 3B (*n* =5)	*p* Value	CKD Stage 2 (*n* = 14)	CKD Stage 3A (*n* = 6)	CKD Stage 3B (*n* = 5)	*p* Value
Pre-vaccination												
Geometric Mean Titers (95% CI)	-	11 (8–15)	11 (8–17)		46 (32–66)	90 (38–210)	53 (33–85)		59 (37–94)	64 (26–153)	61 (23–162)	*0.994* ^a^
Median Titers (Range)	10 (0–40)	10 (10–20)	10 (10–20)	*0.871* ^a^	40 (20–160)	80 (40–320)	40 (40–80)	*0.130* ^a^	60 (20–160)	40 (40–320)	80 (20–160)	*0.470* ^b^
Antibody Titer ≥ 40, *n* (%)	1 (7.1)	0 (0.0)	0 (0.0)	*0.664* ^b^	12 (85.7)	6 (100.0)	5 (100.0)	*0.426* ^b^	11 (78.6)	6 (100.0)	4 (80.0)	
Post-vaccination (1 month)												
Geometric Mean Titers (95% CI)	76 (39–150)	90 (22–369)	40 (22–74)		263 (134–515)	285 (89–914)	184 (51–659)		905 (602–1361)	905 (608–1348)	1114 (758–1637)	*0.571* ^a^
Median Titers (Range)	80 (10–640)	120 (10–320)	40 (20–80)	*0.341* ^a^	240 (40–1280)	320 (80–1280)	160 (40–640)	*0.783* ^a^	960 (320–5120)	960 (640–1280)	1280 (640–1280)	-
Seroprotections, *n* (%)	11 (78.6)	5 (83.3)	4 (80.0)	*0.971* ^b^	14 (100.0)	6 (100.0)	5 (100.0)	-	14 (100.0)	6 (100.0)	5 (100.0)	*0.588* ^b^
Seroconversions, *n* (%)	12 (85.7)	5 (83.3)	5 (100.0)	*0.646* ^b^	10 (71.4)	2 (33.3)	3 (60.0)	*0.281* ^b^	13 (92.9)	5 (83.3)	5 (100)	*0.950* ^a^
Median Fold Titers (95% CI)	8 (2–32)	12 (1–32)	4 (2–8)	*0.344* ^a^	4 (2–16)	2 (1–32)	4 (1–8)	*0.476* ^a^	16 (8–64)	24 (2–32)	16 (8–32)	

Note: ^a^ Kruskal–Wallis test, ^b^ Chi-square test.

## Data Availability

The data presented in this study are available upon request from the corresponding author. The data cannot be accessed publicly as they are restricted for privacy reasons.

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
