# Peer review of "Immunogenicity, Safety, and Efficacy of Influenza Vaccine in T2DM and T2DM with Chronic Kidney Disease"

_vaccines, 2024, doi:10.3390/vaccines12030227_

Round 1

Reviewer 1 Report

Comments and Suggestions for Authors

The main question addressed by the research in the current manuscript was to investigate the efficacy of the influenza vaccine Flu bio in two different patient cohorts, one group diagnosed with Type 2 Diabetes and the other with Type 2 diabetes and who were also diagnosed with chronic kidney disease.  The investigators wanted to know if and demonstrate that there was no significant difference in vaccine responses in patients with T2D and those with T2D-CDK. The vaccine demonstrated a good safety profile.

The study showed that the patients with either T2DM alone or T2DM + CDK were both capable of responding to the influenza vaccine and produced comparable antibody titres. However, a rather glaring omission was that the study did not include a comparison group of aged matched healthy control patients. This would help the investigators to distinguish if patients with T2DM +/- CDK were able to  produce equivalent Ab responses to the flu vaccination compared to healthy controls and if Ab titres in the patient cohorts were at comparable serum levels. This would provide both groups with protection form Influenza infection.

In Tables 3,4 and 5 a measure of seroprotection was and seroconversions was provided- but how were these measured? There was no information provided in the methods about these measures and how they were assessed by the study group.

The data in Table 6 was difficult to interpret as there was no description in the main text to explain the different stages of Stage2, Stage 3A or Stage 3B.  It would be important to define what these different clinical stages represent, and to explain how does it relate to clinical disease progression experienced by the 2DM and/or T2DM +CKD patients or some other? What are the features of the clinical disease at these various stages?

In relation to the virus like illness presented in Figure 1, presumably the investigators were trying to monitor the adverse events in response to the vaccine that was given?  This part of the study needs to be rewritten to make it more clear to the reader. Just monitoring symptom of cold and cough doesn’t seem to be very rigorous – you would imagine these are not common symptoms in response to common vaccines.

It would be good to have seen if the CD8+ T cell response following the flu vaccination increased CTL responses in both T2DM patients groups as control participants.

The discussion is generally focused on restating the results and doesn’t really discuss the main findings of the paper.  In the paragraph from lines 384-389- it talks about IL-6 levels but there is no data provided about IL-6 in the current paper that relates to cytokine measurements.  So its difficult to see how this relates to the current study. IL-6 is an innate cytokine, that can influence B cells growth and differentiation, but what is the relevance for it in this study? The discussion on line 290-294 also does not flow with the remainder of the work- The investigators talk about immune dysregulation in the patients – but there is not evidence that the patients have any level of immune dysregulation as Ab levels between the groups were comparable. This [art of the discussion would need to be reconsidered.

The last paragraph in the discussion also has no direct link to the current study. What is DKD? Is this different clinically from CKD? If so how? What is the relevance of the DKD to the current study?

The references cited were heavily reliant on data sheets from the WHO or clinical guidelines.  The authors should be citing references that relate directly the primary research manuscripts. There appears to be a lack of references that relate to the study of influenza vaccines in patients with T2D or Healthy controls to that matter. As a result the discussion of the manuscript provides a simplistic oversight of the main topic and does not provide much insight of how the current manuscript addresses the gaps in the knowledge. 

Comments on the Quality of English Language

The paper does require editing and spell checking. The methods were appropriate, and the data presented clearly in Tables.  There were typos on Figure 1A that need to be addressed as well.

Need to have a close check on typos.  The word titre, titres was spelt incorrectly throughout.

Line 50 - the sentence should read:  Additional in a previous study...

Line 378 the word strains should be in lowercase.

Author Response

The study showed that the patients with either T2DM alone or T2DM + CDK were both capable of responding to the influenza vaccine and produced comparable antibody titres. However, a rather glaring omission was that the study did not include a comparison group of aged matched healthy control patients. This would help the investigators to distinguish if patients with T2DM +/- CDK were able to produce equivalent Ab responses to the flu vaccination compared to healthy controls and if Ab titres in the patient cohorts were at comparable serum levels. This would provide both groups with protection form Influenza infection.

Thank you for your suggestion. We agree that including healthy controls may help to gain more comprehensive data for antibody titers. However, one of the main objectives of the recent study is to understand the immunogenicity between T2DM and T2DM+CKD. Additionally, since this vaccine already has FDA approval, immunogenicity and safety data for healthy subjects are already established. To reduce the number of participants while still obtaining firm data, healthy participants are not included in this study.

In Tables 3,4 and 5 a measure of seroprotection was and seroconversions was provided- but how were these measured? There was no information provided in the methods about these measures and how they were assessed by the study group.

Thank you for your concern. We defined seroconversion as having antibody titers of at least 1:40 and a fourfold or greater increase in hemagglutination inhibition (HI) titers following vaccination. We recorded subjects from both groups who had antibody titers > 1:40 one month after receiving the vaccine.

This information has been added to the methods section (Lines 147-150).

The data in Table 6 was difficult to interpret as there was no description in the main text to explain the different stages of Stage2, Stage 3A or Stage 3B.  It would be important to define what these different clinical stages represent, and to explain how does it relate to clinical disease progression experienced by the 2DM and/or T2DM +CKD patients or some other? What are the features of the clinical disease at these various stages?

Thank you for your question and  suggestions.

Staging of Chronic Kidney Disease (CKD) is based on the Glomerular Filtration Rate (GFR) in ml/min/1.73m², categorized into G1-G5.

·       Stage 2 or G2 has a GFR of 60-89 ml/min/1.73m².

·       Stage 3A or G3a has a GFR of 45-59 ml/min/1.73m².

·       Stage 3B or G3b has a GFR of 30-44 ml/min/1.73m².

These GFR differences form the basis for CKD staging. Additionally, besides GFR, the determination of CKD staging for patients is also based on the level of proteinuria.

It is already stated in our methods section (Line 111-118)

In relation to the virus like illness presented in Figure 1, presumably the investigators were trying to monitor the adverse events in response to the vaccine that was given?  This part of the study needs to be rewritten to make it more clear to the reader. Just monitoring symptom of cold and cough doesn’t seem to be very rigorous – you would imagine these are not common symptoms in response to common vaccines.

Thank you for your suggestion.

In adults, influenza or flu symptoms typically include experiencing some or all of the following:

·       Fever or feeling feverish/chills

·       Cough

·       Sore throat

·       Runny or stuffy nose

·       Muscle or body aches

·       Headaches

·       Fatigue (tiredness)

The Revision has been completed (Line 376-380)

It would be good to have seen if the CD8+ T cell response following the flu vaccination increased CTL responses in both T2DM patients groups as control participants.

Thank you for your insight. We recognize that this has become a limitation in our study and will take it into consideration for future research (Lines 441-446).

The discussion is generally focused on restating the results and doesn’t really discuss the main findings of the paper.  In the paragraph from lines 384-389- it talks about IL-6 levels but there is no data provided about IL-6 in the current paper that relates to cytokine measurements.  So its difficult to see how this relates to the current study. IL-6 is an innate cytokine, that can influence B cells growth and differentiation, but what is the relevance for it in this study?

The discussion on line 290-294 also does not flow with the remainder of the work- The investigators talk about immune dysregulation in the patients – but there is not evidence that the patients have any level of immune dysregulation as Ab levels between the groups were comparable. This [art of the discussion would need to be reconsidered.

Thank you for your concern. We have rearranged some discussion flow in the discussion section. Additionally, IL-6 does not have a direct relationship with the objectives of this recent study. We have revised the content by removing the discussion on IL-6 as it is not related to the results.

Thank you for your concern. The aims of the recent study are to convey that, despite individuals with DM experiencing immune dysregulation in both cellular and humoral immune systems related to antibody response. We found that the antibody response to B/Washington in the T2DM group is statistically significantly different compared to the T2DM+CKD group (Line 428-434).

The last paragraph in the discussion also has no direct link to the current study. What is DKD? Is this different clinically from CKD? If so how? What is the relevance of the DKD to the current study?

Thank you for your concern.

Diabetic Kidney Disease (DKD) is one of the most common complications of DM, resulting in the occurrence of Chronic Kidney Disease (CKD). Therefore, in the discussion, DKD is briefly mentioned (Line 435-436).

For the diagnosis of DKD, it is based on histopathology. Clinically, DKD with CKD can differ because CKD is a complication of DKD characterized by a decline in kidney function and proteinuria. In contrast, DKD, or alternatively nonproteinuria, does not exhibit proteinuria.

The references cited were heavily reliant on data sheets from the WHO or clinical guidelines.  The authors should be citing references that relate directly the primary research manuscripts. There appears to be a lack of references that relate to the study of influenza vaccines in patients with T2D or Healthy controls to that matter. As a result the discussion of the manuscript provides a simplistic oversight of the main topic and does not provide much insight of how the current manuscript addresses the gaps in the knowledge.

Thank you for your suggestions.

We have made improvements by adding several references related to studies on influenza vaccines in patients with Type 2 Diabetes (T2DM) or healthy controls. (Line 46-49 and Line 60-65)

Comments on the Quality of English Language

The paper does require editing and spell checking. The methods were appropriate, and the data presented clearly in Tables.  There were typos on Figure 1A that need to be addressed as well.

Need to have a close check on typos.  The word titre, titres was spelt incorrectly throughout.

Line 50 - the sentence should read:  Additional in a previous study...

Line 378 the word strains should be in lowercase.

Thank you for your concern. We have read and revised some mistyped and grammatically incorrect sections.

Reviewer 2 Report

Comments and Suggestions for Authors

This is an important and interesting study showing that also T2DM patients with chronic kidney disease develop protective antibody levels against all 3 influenza substrains after trivalent vaccination and that no adverse effects were detected i the trial. 

Major comments: 

1. Please comment on why there are different statistical methods for the analyses, have you addressed normal distribution for each outcome?

2. It is not surprising (or interesting) that the eGFR was different between the two groups when the CK patients were defined based on glomerular filtration rate. 

3. Please merge Table 4, 5 and 6, since the text and result is quite similar and would make the manuscript easier to read. 

4. Please add also "No complains" in Figure 1A for similar appearence as Figure 1B and to make it easier to read. A bit confusing that the text is reffering to Fig 1A, but there is a similar plot in Figure 1B for the numbers (however 34 not 31 without complaints within 1 month). Is figure 1A reffering to accute reactions after vaccination? Please explain.

5. Table 1: Should there be a p value and method chosen for analysis between groups also for BMI Overweight, Pre-obese and Obese?

6. Please explain the categorization of CKD into Stage 2, 3A and 3B

7. Please discuss/explain if the age difference between the two groups could affect antibody levels. 

8: In the abstract, please rephrase the first sentence that "the patients are at high risk of influenza infection.. to something like this: face an increased risk of morbidity and mortality after influenza infection 

Minor comments: 

1. Please change the spelling "titter" to "titer" throughout the manuscript. 

2. In table 3"Median fold titers" p=0.029 is also significant and should be indicated with a star. 

Comments on the Quality of English Language

Ok English, just be aware to change all "titters" to titers throughout the manuscript. 

Author Response

Major comments: 

1. Please comment on why there are different statistical methods for the analyses, have you addressed normal distribution for each outcome?

Thank you for your comment.

The statistical methods used in the research differ due to the presence of both normally distributed and non-normally distributed data. For normally distributed data with a nominal variable scale, the Chi-Square test is employed to examine the difference between two independent samples. Conversely, for non-normally distributed data, the Mann-Whitney test is utilized. The Kruskall-Wallis test method is applied to test the significance of three non-normally distributed samples. Additionally, the Fisher-Exact test is used to assess the hypothesis significance in the comparative analysis of two small independent samples with nominal-shaped data.

This information has been added to the methods section (Line 161-167).

2. It is not surprising (or interesting) that the eGFR was different between the two groups when the CK patients were defined based on glomerular filtration rate. 

Thank you for your comment.

We understand that CKD is defined as abnormalities in kidney structure or function that persist for at least 3 months, impacting health. It is classified based on its cause, Glomerular Filtration Rate (GFR) category, and albuminuria category. 

In this study, we incorporated CKD criteria by examining creatinine and urine albumin levels to estimate GFR (eGFR). The eGFR is calculated using the 2021 CKD-EPI creatinine formula, which includes three variables: age, gender, and creatinine.

3. Please merge Table 4, 5 and 6, since the text and result is quite similar and would make the manuscript easier to read. 

Thank you for your suggestion. We have consolidated the information into one table, namely Table 4.

4. Please add also "No complains" in Figure 1A for similar appearence as Figure 1B and to make it easier to read. A bit confusing that the text is reffering to Fig 1A, but there is a similar plot in Figure 1B for the numbers (however 34 not 31 without complaints within 1 month). Is figure 1A reffering to accute reactions after vaccination? Please explain.

Thank you for your clarification. The figure 1 has been revised accordingly.

Figure 1A, only acute reactions post-vaccine administration are mentioned in both groups. This figure is exclusively dedicated to acute reactions after vaccine administration, as the primary objective is to evaluate the safety of the influenza vaccine.

Figure 1B is designed to illustrate the effectiveness of the influenza vaccine within one month post-vaccine administration. Consequently, it is essential to incorporate the category of "No complaints" in Figure 1B to assess how effectively the influenza vaccine works in both groups.

5. Table 1: Should there be a p value and method chosen for analysis between groups also for BMI Overweight, Pre-obese and Obese?

Obesity is one of the factors leading to chronic low-grade inflammation, ultimately affecting vaccine efficacy.

Table 1 was developed to examine the baseline characteristics of the participants. It is essential to demonstrate that the subjects in both groups have relatively similar characteristics to minimize bias.

6. Please explain the categorization of CKD into Stage 2, 3A and 3B

Based on the KDIGO 2012 Clinical Practice Guideline for the Evaluation and Management of Chronic Kidney Disease, there are five stages of CKD based on Glomerular Filtration Rate (GFR) in ml/min/1.73m², namely G1-G5.  Stage 2 or G2 has a GFR of 60-89 ml/min/1.73m². Stage 3A or G3a has a GFR of 45-59 ml/min/1.73m². Stage 3B or G3b has a GFR of 30-44 ml/min/1.73m².

We have included the references for CKD staging in the methodology section (Line 111-116).

7. Please discuss/explain if the age difference between the two groups could affect antibody levels.

Our recent study did not analyze the age characteristics concerning antibody titers. The discussion does not include an examination of the age correlation with antibody titers in both groups. However, based on the inclusion criteria, the age range for this study was set at 40-59 years. The difference in age was only analyzed in the baseline characteristics to observe distinctions between the two groups.

8. In the abstract, please rephrase the first sentence that "the patients are at high risk of influenza infection.. to something like this: face an increased risk of morbidity and mortality after influenza infection

Thank you for the comment. The revision has been completed

The patients with Type 2 diabetes mellitus (T2DM) and Chronic Kidney Disease (CKD) face an elevated risk of morbidity and mortality following an influenza infection. (Line 21-22)

Minor comments

1. Please change the spelling "titter" to "titer" throughout the manuscript. 

2. In table 3"Median fold titers" p=0.029 is also significant and should be indicated with a star. 

The revision has been completed

The revision has been completed

(line 240-241)